# A Review of Hantavirus Research in Indonesia: Prevalence in Humans and Rodents, and the Discovery of Serang Virus

**DOI:** 10.3390/v11080698

**Published:** 2019-07-31

**Authors:** Nurhayati Lukman, Herman Kosasih, Ima Nurisa Ibrahim, Antonius Arditya Pradana, Aaron Neal, Muhammad Karyana

**Affiliations:** 1Indonesia Research Partnership on Infectious Disease (INA-RESPOND), Jakarta 10560, Indonesia; 2Ima Nurisa Ibrahim, National Institute of Health Research and Development, Ministry of Health of Republic of Indonesia, Jakarta 10560, Indonesia; 3Aaron Neal, National Institute of Allergy and Infectious Disease, National Institutes of Health, Bethesda, MD 20814, USA

**Keywords:** hantavirus, prevalence, Indonesia, Serang virus, human, rodent

## Abstract

Dengue and other common tropical infectious diseases of similar clinical presentation are endemic in Indonesia, which may lead to an underestimation of the prevalence of hantavirus (HTV) infection in the country. To better understand the current burden of HTV infection, this study aimed to both identify acute HTV infection among hospitalized patients with fever and to determine the overall seroprevalence of HTV. These results were further considered within the context of previously reported HTV infection in humans and animals in Indonesia by conducting a review of published literature. As part of an observational cohort study of acute febrile illness, this sub-study retrospectively analyzed blood specimens obtained during admission, during the 2–4-week convalescent period, and three months after admission. Convalescent specimens from patients with clinical signs and symptoms of HTV infection were first screened for HTV IgG. When positive, convalescent specimens and paired acute specimens were screened for HTV IgM, and paired acute specimens were tested for HTV by Reverse Transcription Polymerase Chain Reaction (RT-PCR). A literature review of HTV in Indonesia was conducted on manuscripts manually reviewed for relevance after identification from a search using the terms “hantavirus/Seoul virus” and “Indonesia”. From patients at eight hospitals in seven provincial capitals, HTV IgG seroprevalence was 11.6% (38/327), with the highest being in Denpasar (16.3%, 7/43) and the lowest being in Yogyakarta (3.4%, 1/31). Anti-HTV IgG was most prevalent in adults (13.5%, 33/244) and males (15.6%, 29/186). Acute HTV infections were identified in two subjects, both of whom had Seoul virus. In Indonesia, HTVs have been studied in humans and animals since 1984. Over the past 35 years, the reported seroprevalences in rodents ranged from 0% to 34%, and in humans from 0% to 13%. Fourteen acute infections have been reported, including one in a tourist returning to Germany, but only two have been confirmed by RT-PCR. Almost all rodent and human surveillance results demonstrated serological and molecular evidence of Seoul virus infection. However, in Semarang, anti-Puumala virus IgM has been detected in humans and Puumala RNA in one rodent. In Serang, a new virus named Serang virus was identified due to its differences from Seoul virus. In Maumere, HTV and *Leptospira spp.* were identified simultaneously in rodents. The burden of HTV infection in Indonesia is underestimated, and additional studies are needed to understand the true prevalence. Seroprevalence data reported here, previous observations of HTV co-infections in rodents, and the prevalence of rodent-borne bacterial infections in Indonesia suggest that the population may be routinely encountering HTVs. While Seoul virus appears to be the most prevalent HTV in the country, further studies are needed to understand which HTVs are circulating.

## 1. Introduction

Infectious diseases are the major cause of fever in Indonesia. Since the clinical presentations of common tropical diseases are often difficult to differentiate, and since accurate diagnostic tools are lacking in many healthcare settings, misdiagnosis can easily occur, leading to inappropriate clinical management. As dengue and typhoid fever are predominant in Indonesia, other etiologies such as hantavirus (HTV) infection are often overlooked and rarely diagnosed.

HTVs are rodent-borne single-stranded RNA viruses belonging to the *Bunyaviridae* family. Besides the five widely known HTVs (Hantaan (HTNV), Seoul (SEOV), Puumala (PUUV), Dobrava (DOBV), and Sin Nombre (SNV)), there are 31 other species of HTVs documented by the International Committee on Taxonomy of Viruses [1]. In total, more than 90 HTV genotypes have been identified, of which at least 22 genotypes are pathogenic in humans [2]. Infection with “New World” HTV is generally associated with lungs and can manifest as hantavirus pulmonary syndrome (HPS), while infection with “Old World” HTVs generally affects blood vessels and kidneys and can manifest as hemorrhagic fever with renal syndrome (HFRS) [3].

Only a few acute cases of HTV have been reported in Indonesia, and the only surveillance data available are sporadic and geographically limited [4,5,6,7,8,9,10,11,12]. Given the endemicity of HTV in the country, molecular and serological assays for HTV detection were included in the diagnostic algorithm of a large observational cohort study of patients hospitalized with acute febrile illness (AFIRE). The aim of the HTV sub-study was to estimate the proportion of acute and previous HTV infection in Indonesia amongst hospitalized patients with fever. Our findings are further considered within the context of previously reported HTV infection in humans and animals in Indonesia (Figure 1) by conducting a review of published literature.

## 2. Methods

### 2.1. Estimating the Prevalence of Previous Hantavirus Infection (PHI) and Identifying Acute Hantavirus Infection (AHI)

Patients tested for evidence of PHIs or AHIs were identified from the AFIRE study conducted in Indonesia during 2013–2016 at eight referral hospitals in seven large cities (Figure 1). The study recruited patients who were ≥ 1 year old, hospitalized within the past 24 h with acute fever, and never hospitalized within the past three months. Informed consent was obtained before the collection of clinical data and specimens. Blood was collected from all subjects at enrollment, once during Days 14–28, and at three months after enrollment. Blood and other biological specimens were tested following a diagnostic algorithm, which included the culture of blood, respiratory specimens, and/or urine; microscopic examination of sputum and feces; and molecular and serological screening for dengue virus, *Salmonella typhi, Rickettsia typhi*, *Leptospira spp.*, chikungunya virus, and influenza virus [4].

Convalescent specimens from patients with unidentified etiologies after completing the above testing were screened for anti-HTV IgG by ELISA using Hantavirus IgG DxSelect^TM^ ELISA (Focus Diagnostics, Cypress, CA, USA) following the manufacturer’s instructions. When positive, paired acute and convalescent specimens were tested for anti-HTV IgM using Hantavirus IgM DxSelect^TM^ (Focus Diagnostics, Cypress, USA) and were re-tested for anti-HTV IgG. When serology results indicated AHI, acute specimens were screened by nested reverse transcription PCR using previously published degenerate primers targeting the L segment of HTV [13]. Nested RT-PCR positive HTV specimens were further screened by a previously published species-specific SEOV RT-PCR assay targeting the N gene of the S segment [14]. The SEOV RT-PCR product (250 bp) was sequenced to confirm the identity of Seoul virus.

### 2.2. Review of Published AHIs and PHIs in Humans and Serological and Molecular Evidence of HTV Infections in Rodents

A literature review was conducted on published manuscripts and conference proceedings identified through PubMed, Google Scholar, and Indonesian-language scientific journals and study reports using the keywords “hantavirus” OR “Seoul virus” AND “Indonesia”. The articles were categorized as “AHI”, “surveillance on PHI in humans”, “rodent surveillance”, or a combination of these groups. Each article was reviewed and summarized by five individuals with scientific research backgrounds.

## 3. Results

### 3.1. Prevalence of PHI in Humans

From the 1486 patients enrolled in the AFIRE study, a subset of 327 patients was screened for HTV IgG. Thirty-eight patients were positive, suggesting a PHI prevalence of 11.6%. The prevalence was lowest in Yogyakarta at 3.2% (1/31) and highest in Denpasar at 16.3% (7/43). The prevalence was also found to be higher in adult patients (≥18 years old) at 13.5% (33/244) compared to pediatric patients at 6% (5/83), and significantly higher in male patients compared to female patients, at 15.6% (29/186) and 6.9% (9/131) (*p* value = 0.02 by chi-square test), respectively. The increased prevalence in adults was observed at all sites except Bandung and Makassar, and the increased prevalence in males was observed at all study sites. The prevalence data for PHI is shown in Table 1.

Studies on the prevalence of PHI in the Indonesian population have been conducted since 1991, mostly in coastal areas. The first cross-sectional study recruited 50 subjects working in Maumere harbor, Flores and 50 subjects living in the surrounding area. Using an HTNV-based in-house ELISA 13% (13/100) of patients were positive for anti-HTNV IgG [15] Given that the traffic in ports of Jakarta are busier than those in Maumere, researchers conducted a similar study in 1997 in Sunda Kelapa and Tanjung Priok harbors. Screening patients by the same in-house ELISA identified anti-HTNV IgG in 1.1% (1/92) and 1.8% (2/113) of subjects, respectively [16]. Based on these results, a community study was conducted in 2000 targeting the areas where suspected HTV patients lived. Sera from 986 subjects were tested for anti-HTV IgG by a commercial ELISA that used recombinant SEOV and SNV antigens (Focus Technologies, Cypress, CA, USA). This study observed prevalences of 0% (0/176) in Serang, 0.67% (2/299) in Semarang, 0.8% (2/249) in Subang, and 3.1% (8/262) in Jakarta [11]. Using the same commercial ELISA kits, another cross-sectional study was conducted in 2004 among 85 patients with kidney injury in Makassar and Jakarta identifying a prevalence of 8.2% [17]. The most recent serosurvey conducted in 2009 in the Thousand Islands, north of Jakarta did not find any anti-HTV IgG among 51 sera screened by HTV-based IFA [9]. The ports or districts where studies were conducted, laboratory methods, and results are shown in Figure 1.

### 3.2. Acute Hantavirus Infections

Among all published literature, only fourteen AHIs have been reported in Indonesia to-date [4,5,6,7,8]. The first report described 10 AHIs identified retrospectively through serological screening of sera from 94 confirmed non-dengue febrile patients visiting two hospitals in Semarang and Yogyakarta during 1995–1996. Serial serum specimens were tested for anti-HTV IgM using a commercial ELISA based on recombinant SEOV and SNV nucleocapsid protein (Focus Technologies, Cypress, CA, USA). Positive ELISA results were confirmed by immunofluorescence assay (IFA) using SEOV, HTNV, and PUUV serotypes. Sera were also tested for anti-HTV IgG by ELISA, IFA, and immunoblotting. All 10 cases were IgM positive by IFA, five cases were IgG positive by IFA, and a four-fold increase in IgM titers was detected in three of the cases. Interestingly, the IFAs for all 10 patients showed strong reactivity with the PUUV serotype, with titers between 64 and ≥256, and to a lesser extent with HTNV and SEOV serotypes, with titers between 16 and 128. RT-PCR was not performed on any of the acute specimens. The etiologies of the remaining 84 cases included *Rickettsia typhi* (5), rubella virus (3), *Orientia tsutsugamushi* (2) chikungunya virus (2), *Leptospira spp.* (2), and influenza A virus (1) [8].

A second report of AHI was identified from a hospital-based study in Bandung conducted from 2004 to 2005, which screened subjects with fever accompanied by at least one of the following: hemorrhagic manifestations, thrombocytopenia, liver dysfunction, renal insufficiency or non-cardiogenic pulmonary edema. Among 406 subjects, only one subject showed four-fold increases in anti-HTV IgM and IgG by commercial ELISA (Focus Technologies, Cypress, CA, USA), and in-house ELISA based on HTNV cell-lysate antigen. RT-PCR was negative for HTV, though anti-SEOV IgG and SEOV RNA were detected in rodents trapped around the house of the patient [6].

A third report identified an AHI in a 70-year-old traveler returning to Germany from a recent visit to Sulawesi, Indonesia in 2017 [7]. This patient was hospitalized with acute kidney injury, three weeks after the onset of fever. Serum screened by immunoblot assay showed strong reactivity against DOBV and HTNV, and weak reactivity against SEOV though partial sequences of S and L genes grouped with the SEOV serotype (GenBank MG386253 and MG386252). The most recent report of AHIs described two cases, one from Jakarta and one from Surabaya, observed during the AFIRE study. Neither patient was diagnosed with HTV infection during hospitalization, but AHIs were retrospectively confirmed through sero-conversion or four-fold increase of anti-HTV IgM and IgG, and SEOV RNA [4]. Demographic details, methods of confirmation, and clinical and laboratory findings are summarized in Table 2.

In all 14 AHI cases, with the exception of the returning traveler, patients were initially suspected to have dengue, which was ultimately ruled out. Thrombocytopenia (platelets <150,000 cells/dL) was reported in two cases from Central Java and in all of the cases from Bandung, Jakarta, and Surabaya. Rash was reported in all five patients with documented data from Central Java, and icterus with very high levels of transaminase was found in one patient. Unlike the 13 Indonesian patients, whose creatinine levels were within normal limits, the German man was hospitalized for acute kidney injury (creatinine level >3.7 mg/dL) after an unusually protracted clinical course, which also included diarrhea, a symptom that was not reported in the Indonesian patients. The German AHI was the only case successfully diagnosed as HTV infection during hospitalization.

### 3.3. Serological Surveillance in Rodents

Rodent surveillance to identify HTV infection started in 1984, seven years earlier than human surveillance in Indonesia. Specimens from rodents collected in Makassar and Semarang showed seroreactivity against HTNV but exact numbers are unavailable [18]. To clarify this finding, rodent specimens collected in several ports in Indonesia from 1985 to 1992 were tested retrospectively using archived specimens for anti-SEOV antibodies using IFA and when positive an immune adherent hemagglutination test. From 194 rodent specimens (*Rattus norvegicus* (164), *R. tanezumi* (10), *R. rattus diardii* (14), *R. exulans* (4), and *R. tiomanicus* (2)) collected in Makassar 25 (12.9%) were positive by IFA, 24 of which were from *R. norvegicus*. Specimens from the areas of Pasuruan, Maumere, Panjang (Bengkulu), Cilacap, and Jakarta showed no evidence of HTV infection, and specimens from Semarang had a prevalence of only 0.5% [10]. In contrast, a different group reported an anti-HTNV IgG prevalence of 10.2% (19/186) in *R. tanezumi* in Maumere in 1991 [15]. Given these diverse results and considering that the two harbors of Tanjung Priok and Sunda Kelapa in north Jakarta handle national and international shipping, further rodent surveillance was conducted there in 1995–1996. The prevalence of anti-HTNV IgG in 1995 was 9.1% (3/33) and 20.3% (12/59), respectively [18]. In 1996, the study was expanded into the neighborhoods around the harbors, identifying prevalences of 33% (37/112) and 21.3% (19/89), respectively [16]. During this two-year study, HTV antibody prevalence was highest in *R. norvegicus* at 28.4% (54/190), *R. tanezumi* at 17.4% (16/92) and *Mus musculus* at 100% (1/1). Anti-HTNV IgG was also detected in 37.5% (3/8) of *Suncus murinus* trapped [16,18]. In 1999, rodent surveillance included Batam, a harbor close to Singapore and Makassar. The prevalence of HTNV infections was 2.3%, and 5.1%, respectively [19]. In 2000, a community study was conducted in four areas on Java targeting neighborhoods of patients suspected to have HTV infections. Anti-HTV IgG was examined by commercial ELISA based on recombinant SEOV and SNV antigens (Focus Technologies, Cypress, CA, USA). The reported prevalences of anti-HTV IgG were 3.1% (2/64) in Subang, 7.5% (7/93) in Serang, 9.4% (13/138) in Semarang, and 23.7% (18/76) in Central Jakarta [11]. In 2005, human AHI index case was confirmed by a four-fold anti-HTV IgM/IgG increase prompted a rodent surveillance study concentrating on rodents collected from around the homes of the AHI case and a non-case control. From those two areas, anti-HTV IgG was found in 13.2% (21/159) and 4.7% (4/86) of rodents, respectively [6]. Most recently, a study conducted in the Thousand Islands area north of Jakarta revealed an increasing anti-HTV antibody prevalence of 15.9% (27/170) in 2005 and 33.9% (20/59) in 2009 [9]. The surveillance sites, laboratory methods, and results are shown in Figure 1.

### 3.4. Molecular Surveillance in Rodents

The first molecular surveillance of rodents in Indonesia occurred in 2000 in four communities across four different areas of Java. From collected rodents that were positive for anti-HTV IgG, and RT-PCR was positive in 52.5% (21/40). The highest prevalence was seen in the community of Cempaka Baru, Jakarta, where 11 of 18 rodents were found to have SEOV RNA. In Semarang, 8 of 13 rodents had HTV RNA (seven SEOV and one PUUV). In Serang, two of seven rodents also had HTV RNA (one SEOV and one PUUV). In Subang, HTV RNA was not detected in two rodents with anti-HTV IgG [11]. A separate group further investigated lung tissue collected from one of the SEOV-positive Serang rodents. The tissue was re-screened using HTV genus-specific L segment primers and sequenced, revealing substantial diversity in the nt 2956–3367 sequence (GenBank: AM998806.1). This result suggested the presence of a novel HTV, provisionally named Serang virus (SERV), which was most closely related to SEOV carried by *R. norvegicus* and Thailand virus carried by *Bandicota indica* [20]. Additional comparisons of the partial S segment sequence (nt 29–862) and the M segment sequence (nt 1970–2332) further corroborated the results of L segment sequence analysis (GenBank: AM998808.1, and AM998807.1, respectively).

In 2005, as part of the Bandung case and non-case control study mentioned above, 17 rodents with anti-HTV IgG were further screened by RT-PCR. SEOV was identified in 13 (76.5%) rodents, primarily from lung, spleen, and kidney, but not liver specimens. Sequence analysis of a 225-bp region of the S segment revealed 99% identity with SEOV isolated from *R. norvegicus* in Singapore in 2006 [6]. In the above-mentioned 2009 Thousand Islands study, three lung tissue specimens were examined for the presence of HTV RNA. Results from RT-PCR and sequencing of the partial S and M segments revealed a SEOV cluster with isolates detected in Vietnam in 2007 and Singapore in 2006 [9].

The most recent molecular surveillance occurred in Maumere in 2014, where 114 lung tissue specimens were screened by RT-PCR for HTV using L segment primers. Twenty-six (22.8%) of the rodents were positive, but further speciation was not conducted. Kidney tissues from the same rodents were also tested for *Leptospira spp.* using primers specific for the 16S rRNA gene, identifying five (4.4%) positives. Interestingly, two (1.8%) rodents were positive for both HTV and *Leptospira spp.* [12].

## 4. Discussion

HTV infection is a rarely studied disease in Indonesia, though our human seroprevalence study and HTV literature review identified several important findings to guide the future of HTV research in the country. First, the seroprevalence of HTV IgG in our study was higher than in several previous studies, and the populations most affected were men, adults, and those with kidney injury. Second, despite the high prevalence of PHIs, AHIs were infrequent and difficult to distinguish from other tropical diseases, particularly dengue. Third, SEOV was the most common species of HTV identified in Indonesia, although it is clear that other species are also circulating in the region.

Overall, our study during 2013–2016 and a cross-sectional surveillance in Maumere in 1991 identified higher prevalences of PHI (11.6% and 13%, respectively) compared to several previous studies, ranging from 0% in Thousand Islands to 8.2% among patients with kidney injury in Jakarta and Makassar. One possible explanation for the large differences may be the diagnostic assays used since the various IFAs and ELISAs differed in their specific HTV targets. However, it has been demonstrated that HTNV-based assays are cross-reactive with IgM/IgG against SEOV [21], the predominant HTV detected in Indonesia, thus the differences may be better explained by the sample sizes and population across the studies. Our serosurvey of 327 patients is the second largest conducted in Indonesia to-date and captures a representative cross-section of hospitalized patients in major urban areas. Previous studies focused primarily on healthy populations in coastal regions and harbors, and it is possible that HTV infection may be lower in those regions compared to large urban centers. Alternatively, the population in our study may not be truly representative of the general population. From the AFIRE study, 151/1486 (10.2%) acute febrile patients tested positive for either *Rickettsia typhi* or *Leptospira spp.* [22], suggesting that our AFIRE population may be at a generally increased risk of exposure to other rodent-borne pathogens, including HTV. Lastly, and most alarmingly, the observed differences in PHI prevalence could be due to increased HTV transmission over time. In Semarang alone, our observed prevalence of 13.4% (9/67) is significantly higher than the previously reported prevalence of 0.8% (2/249) in 2000. Since these studies are not directly comparable, a broad serosurvey across several communities and populations is necessary to understand the true burden of HTV in Indonesia. Our sub-study identified that men and adults are the populations with the highest seroprevalence. Men in Indonesia may be at greater risk through general exposure to rodents during work and outdoor activities [23], whereas the seroprevalence seen in adults may be explained by an accumulated lifelong exposure to HTV and reservoirs since anti-HTV IgG may persist for a long time [24]. The higher seroprevalence seen in Makassar and Jakarta in those with kidney injury is understandable since HTV infection can damage the tubules of the kidney [3], causing the symptoms and signs of kidney disease as seen in the AHI in the returning traveler [7].

AHI remains rarely identified in Indonesia, though its prevalence is likely underestimated. In all reported cases, except in the case of the German tourist, AHI was found through the initial dengue screening. Dengue was likely suspected due to the shared clinical signs and symptoms of skin rash, hemorrhagic manifestations, and thrombocytopenia. The typical HTV sign of acute kidney injury, measured by increased creatinine level, may not be a helpful variable in differentiating dengue and HTV given that the sign was only seen in the German tourist. The typical SEOV sign of liver involvement [25], measured by elevated liver transaminases, was also not reported in all of the cases from Semarang and Yogyakarta. Leukopenia may be a helpful distinguishing variable, as it has been reported in 79% of dengue cases [26] and in only half of the 14 Indonesian AHI cases. The development of an accurate, rapid dengue NS1 test would be a valuable tool for quickly determining a patient’s dengue status so that when negative, clinicians could more confidently consider AHI.

SEOV was the predominant serotype identified in both rodents and humans based on molecular assays. This is not surprising given the prevalence of its major vectors, *R. norvegicus* and *R. tanezumi* throughout the study areas and Indonesia in general. However, the detection of PUUV in rodents from Semarang and Serang and the strong reactivity of sera from patients in Semarang and Yogyakarta against PUUV is interesting since PUUV is only known to be carried by the bank vole *Myodes glareolus* [3], which is not found in Indonesia. Interestingly, a PUUV-like virus was found in grey voles in China [27], suggesting that the host range of this serotype may be poorly understood. Antibody cross-reactivity among HTV serotypes particularly between HTNV and SEOV [21] may explain the seroreactivity to PUUV or HTNV in the absence of PUUV or HTNV RNA and the presence of SEOV RNA commonly seen in rodents. Given the potential for antibody cross-reactivity, and that PUUV aplicons were not sequenced, the potential existence of PUUV in Indonesia requires further examination. The detection of anti-HTNV antibodies in 3 *S. murinus* shrews suggests that common HTVs may also be harbored by non-rodent mammals. This is not unprecedented since the HTV Thottapalayam virus has been reported to circulate in shrews [28].

Despite limited research on HTVs in Indonesia, the discovery of Serang virus is significant and should encourage investigation into the potential diversity of HTVs in Indonesia and their potential for recombination and cross-species transmission [20].

HTVs are clearly endemic in Indonesia, although they are often overlooked due to the similar clinical manifestations they share with dengue, the most common tropical disease in the country, and the unavailability of reliable diagnostic tests at health care centers. An integrative prospective study involving a broadly-representative population of humans, rodents, and environments across the diverse geographical areas and seasons of Indonesia is needed to develop a better understanding of the epidemiology clinical burden, and transmission of this important disease.

## Figures and Tables

**Figure 1 viruses-11-00698-f001:**
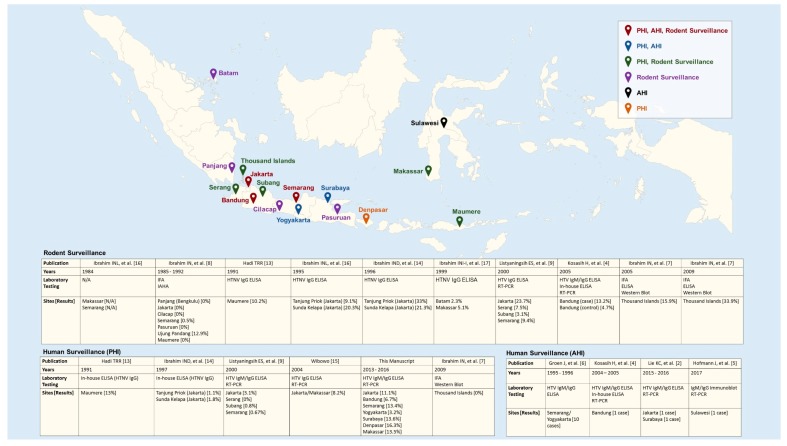
The surveillance sites, laboratory methods, and results.

**Table 1 viruses-11-00698-t001:** The prevalence of previous hantavirus infection by AFIRE study sites, 2013–2016.

	Tested/ Subjects	Positive (%)	Pediatric Pos/Tested (%)	Adults Pos/Tested (%)	Male Pos/Tested (%)	Female Pos/Tested (%)
Bandung	45/269	3 (6.7)	1/12 (8.3)	2/33 (6.1)	2/27 (7.4)	1/18 (5.6)
Jakarta	45/156	5 (11.1)	2/25 (8)	3/20 (15)	3/25 (12)	2/20 (10)
Semarang	67/257	9 (13.4)	0/7 (0)	9/60 (15)	6/35 (17.1)	3/32 (9.4)
Yogyakarta	31/169	1 (3.2)	0/10 (0)	1/21 (4.8)	1/20 (5)	0/11 (0)
Surabaya	44/221	6 (13.6)	0/9 (0)	6/35 (17.1)	6/23 (26)	0/21 (0)
Denpasar	43/213	7 (16.3)	0/7 (0)	7/36 (19.4)	6/23 (26)	1/20 (5)
Makassar	52/201	7 (13.5)	2/13 (15.4)	5/39 (12.8)	5/33 (15.2)	2/19 (10.5)
Total	327/1486	38 (11.6)	5/83 (6)	33/244 (13.5)	29/186 (15.6)	9/131 (6.9)

**Table 2 viruses-11-00698-t002:** Demographics, methods, clinical characteristics, and laboratory findings from reports of acute hantavirus infections in Indonesia.

	Groen (2002), Suharti (2009)	Kosasih (2011)	Hofmann (2018)	Lie (2018)
City/island	Semarang, Yogyakarta (Central Java)	Bandung	Sulawesi	Surabaya, Jakarta
Year(s)	1995–1996	2004–2005	2017	2015–2016
Number of cases	10	1	1	2
Methods of confirmation	HTV IgM and IgG ELISA and IFA (SEOV, HTN, PUU), Immunoblotting	HTV IgM and IgG ELISA, in-house ELISA RT-PCR (negative)	IgM, IgG Immunoblot (DOBV, HTNV, SEOV), RT-PCR (positive)	HTV IgM and IgG ELISA RT-PCR (positive)
Evidence of rodent infection	N/A	Yes	N/A	N/A
**Demography ***				
Age	Mean: 17.6 (13-N/A) 90% under 20 years old	25	70	55; 27
Gender (male:female)	2:8	male	male	1:1
**Clinical ****				
Constitutional symptoms	100% (Fever, headache) 80% (Weakness, joint/muscle pain, nausea/vomiting, epigastric pain) 20% (Retro-orbital pain)	Fever, headache, nausea, vomiting, abdominal discomfort, loss of appetite, malaise	Fever, severe diarrhea, thoracic/back pain, bronchopulmonary symptom	Patient 1. Fever, abdominal/epigastric pain, nausea, vomiting, retro-orbital pain. Patient 2. Fever, anorexia, headache, nausea, vomiting, arthralgia
Signs	40% Hepatomegaly 20% (Lymphadenopathy, epistaxis/gum bleeding, hematemesis/melena)			Patient 1. Lethargy Patient 2. Decrease of consciousness, jaundice.
rash	40% Exanthema 80% Petechiae			
Hematology (reference range)	Median (range)			
Hb (mg/dL) **	12 (10.6–13.7)	N/A	17.6	13.2; 16.5
Hct (40–50%) **	37.7 (34–41.7)	50	N/A	38.8; 50.6
Leukocytes (3500–9000 cells/dL) *	Normal leukocytes (50%) 5300 (3600–6300)	9600	11.7–16.8	3850; 5790
Platelet (150,000–450,000 cells/dL) *	Normal Thrombocytes (50%) 155,000 (24,000–267,000)	38,000	66,000	50,000; 24,700
Lymphocyte (23.1–49.9%)	N/A	N/A	N/A	8.4; 16
**Chemistries ****				
BUN (10–23 mg/dL)	N/A	25	240	10; 20.3
Creatinine (0.5–1.2 mg/dL)	0.8 (0.6–0.93)	0.9	5.5	0.57; 1.16
Protein (6.3–8.2 g/dL)	N/A	5.8	N/A	N/A
Albumin (3.5–5.2 g/dL)	N/A	3.3	N/A	N/A
Bilirubin (<1.4 mg/dL)	0.53 (0.46–1.32)	N/A	N/A	0.97; 4.68
SGOT (<38 U/L)	15 (3–56)	130	234	216; 3900
SGPT (<41 U/L)	17 (6–27)	76	173	116; 891
GGT (<60 U/L)	N/A	N/A	159	N/A
CRP (<5 mg/dL)	N/A	N/A	54.4	N/A; 41.1
Procalcitonin (<0.5 ng/dL)	N/A	N/A	N/A	N/A; 5.98

* from 10 subjects, ** from five subjects.

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
