# Peer review of "A Review of Hantavirus Research in Indonesia: Prevalence in Humans and Rodents, and the Discovery of Serang Virus"

_viruses, 2019, doi:10.3390/v11080698_

Round 1
Reviewer 1 Report
Line 60. Delete “the” or add “febrile illness”
Lines 62-64. There are many more hantaviruses than the five mentioned here. See Ecohealth. 2018 Mar;15(1):163-208. doi: 10.1007/s10393-017-1305-2. Epub 2018 Apr 30. Global Diversity and Distribution of Hantaviruses and Their Hosts.
Milholland MT1, Castro-Arellano I2, Suzán G3, Garcia-Peña GE3,4,5, Lee TE Jr6, Rohde RE7, Alonso Aguirre A8, Mills JN9. This reference indicates that there are over 90 hantavirus genotypes, 22 of which are known human pathogens.
Table 1. and discussion. Is the difference in antibody prevalence between men and women statistically significant? Has the difference been tested for significance? If not, it should be. If not significant the discussion should be modified to reflect that. If so, say so.
Lines 153-163. Detection of PUUV is indeed surprising. A distinct, PUUV-like virus was found in the grey vole in Cina (Zhang et al. 2007 J Med Virol 79: 1208-12180). Might the PUUV virus found by the authors similarly be a variant or even genetically distinct virus related to PUUV?
Author Response
Response to Reviewer 1 Comments
Reviewer 1:
Thank you very much for your thorough review.
Point 1: Line 60. Delete “the” or add “febrile illness”
Response 1: ‘The’ has been deleted from the text.
Point 2: Lines 62-64. There are many more hantaviruses than the five mentioned here. See Ecohealth. 2018 Mar;15(1):163-208. doi: 10.1007/s10393-017-1305-2. Epub 2018 Apr 30. Global Diversity and Distribution of Hantaviruses and Their Hosts. Milholland MT1, Castro-Arellano I2, Suzán G3, Garcia-Peña GE3,4,5, Lee TE Jr6, Rohde RE7, Alonso Aguirre A8, Mills JN9. This reference indicates that there are over 90 hantavirus genotypes, 22 of which are known human pathogens.
Response 2: We agree with the reviewer. The text has been updated and now also includes the number of hantavirus species that has been listed in the International Committee on Taxonomy of Viruses. The text now reads (line 62-65):
“HTNs are rodent-borne single-stranded RNA viruses belonging to the Bunyaviridae family. Besides the five widely known HTVs (Hantaan (HTNV), Seoul (SEOV), Puumala (PUUV), Dobrava (DOBV), and Sin Nombre (SNV)), there are 31 other species of HTVs documented by the International Committee on Taxonomy of Viruses. In total, more than 90 HTV genotypes have been identified, of which, at least 22 genotypes are pathogenic in humans.”
In addition, the information, “each with their own specific reservoirs and geographic distributions,” has been removed since Milholland et al. (2018) stated that two or more host species may share a hantavirus, and two or more hantaviruses may share one host. Furthermore, the authors highlight the need to re-evaluate the one virus-one host paradigm.
Point 3: Table 1 and discussion. Is the difference in antibody prevalence between men and women statistically significant? Has the difference been tested for significance? If not, it should be. If not significant the discussion should be modified to reflect that. If so, say so.
Response 3: The prevalence is men is significantly higher than in women (p=0.02). This information has been added to the results and discussion sections, which now reads:
Line 115-116: “The prevalence was also found to be higher in adult patients (≥18 years old) at 13.5% (33/244) compared to pediatric patients at 6% (5/83), and significantly higher in male patients compared to female patients, at 15.6% (29/186) and 6.9% (9/131) (p value=0.02 by chi-square test), respectively.”
Point 4: Lines 153-163. Detection of PUUV is indeed surprising. A distinct, PUUV-like virus was found in the grey vole in China (Zhang et al. 2007 J Med Virol 79: 1208-12180). Might the PUUV virus found by the authors similarly be a variant or even genetically distinct virus related to PUUV?
Response 4: We are unable to comment directly on the similarities and differences of the Puumala viruses found in China (Zhang et al., 2007) and in the previous study by Listiyaningsih (2005). The study by Zhang et al. used primers for PUUV from S and M segments, followed by sequencing, whereas Listiyaningsih et al. used a different primer for the S segment of PUUV (referring to Dekonenko et al. 1997) and did not perform sequencing. For this reason, and the potential for antibody cross-reactivity, the potential existence of PUUV in Indonesia requires further examination, as stated in the manuscript.
We added to the discussion sections, which now reads:
Line 156-157: Interestingly, a PUUV-like virus was found in grey voles in China (27), suggesting that the host range of this serotype may be poorly understood.
Reviewer 2 Report
This study contributes to the awareness of hantavirus infection in Indonesia. Although it is a retrospective study and the study population may not be representative for the general population conclusions are just indicative, but results show that there is a a reason and there is a need in both public health and clinical perspective to raise awareness under professionals. It's good to mention in the conclusions that the next step will be a prospective study including a representative population studying both the human, the veterinarian and environmental aspects to learn more about epidemiology , transmission and transmission risks.
Author Response
Point 1: This study contributes to the awareness of hantavirus infection in Indonesia. Although it is a retrospective study and the study population may not be representative for the general population conclusions are just indicative, but results show that there is a reason and there is a need in both public health and clinical perspective to raise awareness under professionals. It's good to mention in the conclusions that the next step will be a prospective study including a representative population studying both the human, the veterinarian and environmental aspects to learn more about epidemiology, transmission and transmission risks.
Response 1: Thank you very much for your review.
The following text has been added to the conclusions/recommendation section (line 165-173):
“Despite limited research on HTVs in Indonesia, the discovery of Serang virus is significant and should encourage investigation into the potential diversity of HTVs in Indonesia and their potential for recombination and cross-species transmission (20).
HTVs are clearly endemic in Indonesia, though they are often overlooked due to the similar clinical manifestations they share with dengue, the most common tropical disease in the country, and the unavailability of reliable diagnostic tests at health care centers. An integrative prospective study involving a broadly-representative population of humans, rodents, and environments across the diverse geographical areas and seasons of Indonesia is needed to develop a better understanding of the epidemiology, clinical burden, and transmission of this important disease.